# SARS-CoV-2 infections among Australian passengers on the Diamond Princess cruise ship: A retrospective cohort study

Liz J. Walker[1,2], Tudor A. Codreanu[3,4], Paul K. Armstrong[3,4], Sam Goodwin[3,5], Abigail Trewin[4], Emma Spencer[5], Samantha M. Colquhoun[2], Dianne M. Stephens[3,5], Rob W. Baird[6], Nicholas M. Douglas[6,7], Danielle Cribb[1], Rhonda Owen[1], Paul Kelly[1,2], Martyn D. Kirk[1,2]*

1 Australian Government Department of Health, Canberra, Australian Capital Territory, Australia, 2 National Centre for Epidemiology and Population Health, The Australian National University, Canberra, Australian Capital Territory, Australia, 3 National Critical Care and Trauma Response Centre, Darwin, Northern Territory, Australia, 4 Western Australian Department of Health, Perth, Western Australia, Australia, 5 Northern Territory Department of Health, Darwin, Northern Territory, Australia, 6 Territory Pathology, Department of Health, Darwin, Northern Territory, Australia, 7 Global and Tropical Health Division, Menzies School of Health Research, Charles Darwin University, Darwin, Australia

* martyn.kirk@anu.edu.au

**Data Availability Statement:** The dataset used in the manuscript 'SARS-CoV-2 infections among Australian passengers on the Diamond Princess

## Abstract

### Background

Prolonged periods of confined living on a cruise ship increase the risk for respiratory disease transmission. We describe the epidemiology and clinical characteristics of a SARS-CoV-2 outbreak in Australian passengers on the Diamond Princess cruise ship and provide recommendations to mitigate future cruise ship outbreaks.

### Methods

We conducted a retrospective cohort study of Australian passengers who travelled on the Diamond Princess from 20 January until 4 February 2020 and were either hospitalised, remained in Japan or repatriated. The main outcome measures included an epidemic curve, demographics, symptoms, clinical and radiological signs, risk factors and length of time to clear infection.

### Results

Among 223 Australian passengers, 56 were confirmed SARS-CoV-2 positive. Forty-nine cases had data available and of these over 70% had symptoms consistent with COVID-19. Of symptomatic cases, 17% showed signs and symptoms before the ship implemented quarantine and a further two-thirds had symptoms within one incubation period of quarantine commencing. Prior to ship-based quarantine, exposure to a close contact or cabin mate later confirmed SARS-CoV-2 positive was associated with a 3.78 fold (95% CI, 2.24–6.37) higher risk of COVID-19 acquisition compared to non-exposed passengers. Exposure to a positive cabin mate during the ship's quarantine carried a relative risk of 6.18 (95% CI,

cruise ship' contains demographics, symptoms, clinical and radiological signs, risk factors and length of time to clear infection for 223 Australians and is held in a REDCap database. The dataset was collected under the Health Security Act (2005). The dataset is restricted to ensure participants privacy and confidentiality. These restrictions are imposed by the Australian National University Human Research Ethics Committee (Protocol 2017/909) and the Australian Government Department of Health. Inquiries about data access can be sent to the Assistant Secretary, Health Emergency Management Branch, Australian Government Department of Health, health.ops@health.gov.au.

**Funding:** The author(s) received no specific funding for this work.

**Competing interests:** The authors have declared that no competing interests exist.

1.96–19.46) of developing COVID-19. Persistently asymptomatic cases represented 29% of total cases. The median time to the first of two consecutive negative PCR-based SARS-CoV-2 assays was 13 days for asymptomatic cases and 19 days for symptomatic cases ($p = 0.002$).

## Conclusion

Ship based quarantine was effective at reducing transmission of SARS-CoV-2 amongst Australian passengers, but the risk of infection was higher if an individual shared a cabin or was a close contact of a confirmed case. Managing COVID-19 in cruise ship passengers is challenging and requires enhanced health measures and access to onshore quarantine and isolation facilities.

## Introduction

Coronavirus disease 2019 (COVID-19) is a viral respiratory infection caused by the severe acute respiratory syndrome coronavirus 2 (SARS-CoV-2). First detected in Wuhan, China in early December 2019, the virus spread quickly resulting in a global pandemic with substantial morbidity and mortality. Common symptoms of infection include fever (>37.5˚C), cough, fatigue, and a sore throat [1, 2]. Asymptomatic cases account for a minority of infections [3] but are implicated in the transmission of SARS-CoV-2 [4].

On 20 January 2020, the Diamond Princess cruise ship, left Yokohama, Japan on a 14-day tour with shore excursions in Japan, Hong Kong, Vietnam and Taiwan. Five days after departure, a passenger with a two-day history of mild dry cough disembarked in Hong Kong and was later confirmed SARS-CoV-2 positive [5, 6]. Four weeks later, the largest outbreak of SARS-CoV-2 outside of China had occurred on board, ultimately infecting 19% (712/3,711) of passengers and crew and causing 13 deaths [7, 8]. In response to the outbreak, Japanese authorities implemented a 14-day quarantine from 5 February, requiring all passengers to remain in their cabins, apart from periodic supervised time in outdoor public areas for exercise and well-being purposes. Symptomatic and asymptomatic passengers with laboratory confirmed COVID-19 were disembarked and transported to isolation wards in Japanese hospitals [5]. Crew deemed asymptomatic continued to performed essential services throughout the ship [5].

Of the 3,711 passengers and crew on board the Diamond Princess, 223 (6%) were Australian citizens. Following two weeks of ship-based quarantine, on 19 February the Australian Government responded by repatriating 156 asymptomatic Australians, six foreign partners and four crew to a quarantine facility located at Howard Springs, Darwin, Northern Territory, where they were quarantined for a further 14 days. In-country medical health liaison support was also provided to 46 infected Australians who remained in Japanese hospitals and 21 SARS-CoV-2 negative relatives.

The Diamond Princess outbreak is of global interest and has been pivotal in understanding transmission dynamics of SARS-CoV-2, changes to the reproductive factor of SARS-CoV-2 in various stages of quarantine, predominant disease symptoms and the role of asymptomatic cases. Descriptive analyses and case series of the Diamond Princess outbreak have previously been conducted, however to our knowledge this is the first study to describe the epidemiology

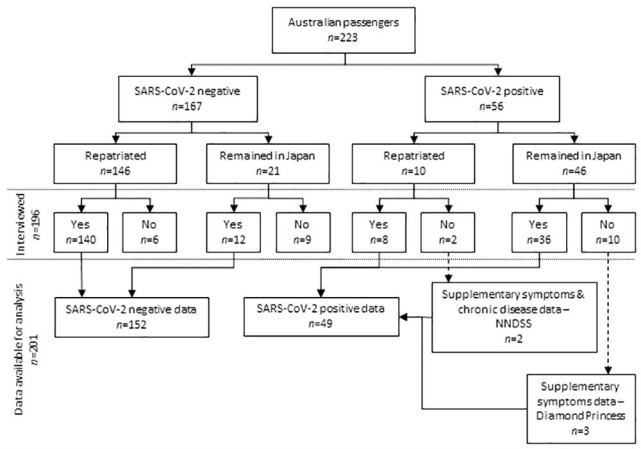

**Fig 1. Flowchart of study recruitment and analysis of Australian passengers from the Diamond Princess cruise ship, March 2020.**

and clinical characteristics of a SARS-CoV-2 outbreak among Australian passengers and provide recommendations to mitigate future outbreaks.

## Methods

This retrospective cohort study of Australian passengers on board the Diamond Princess cruise ship was conducted through face-to-face or telephone interviews, in Australia and Japan, between 1 March and 6 April 2020. We identified five additional SARS-CoV-2 positive cases from the Australian National Notifiable Disease Surveillance System (NNDSS) and the Diamond Princess manifest (Fig 1).

Data were collected using a paper-based questionnaire modified from the World Health Organization's 'The First Few X cases and contacts (FFX) investigation protocol for coronavirus disease 2019 (COVID-19)' [9]. We sought additional information about the passengers' accommodation, shore excursion attendance and movement on board the vessel before and during the ship-based quarantine. The majority of interviews were conducted in the quarantine facility at Howard Springs or via telephone for people remaining in Japan. Questionnaires were completed within two weeks of disembarkation to reduce the risk of recall bias. A RED-CAP database (REDCap 9.1.3 © 2020 Vanderbilt University) was used for data entry and secure storage.

We used the confirmed SARS-CoV-2 case definition as provided by the Communicable Disease Network Australia, Series of National Guidelines (SoNG) Coronavirus Disease 2019 (COVID-19), version 1.8 [10], that is, a positive reverse transcriptase polymerase chain reaction (RT-PCR) for SARS-CoV-2, with or without symptoms, on an oropharyngeal or nasopharyngeal specimen. RT-PCR tests performed in Japan followed a protocol endorsed by the Japanese National Institutes of Infectious Diseases [1]. The diagnosis of repatriates in Darwin, Australia included an in-house RT-PCR assay conducted on a Roche™ (Switzerland) Lightcycler with probes sourced from Sigma-Aldrich™ (USA) targeting ORF1 and S genes, and a multiplexed-tandem PCR assay targeting ORF1 from AusDiagnostics™ (Australia). The Victorian Infectious Diseases Reference Laboratory in Melbourne, Australia confirmed the results, with no discrepancies found. Testing of repatriates at the Howard Springs facility was done on the basis of symptoms. No asymptomatic testing was conducted in Australia following

repatriation. At the time of the study, anosmia (loss of smell) and/or ageusia (loss of taste) were not recognised symptoms of SARS-CoV-2 infection. We defined non-specific acute respiratory illness (ARI) as at least one of the following signs and symptoms occurring after embarkation on the Diamond Princess or repatriation, fever, cough, shortness of breath, sore throat, runny nose, headache or unusual fatigue, in a person who had not tested positive to either SARS-CoV-2 or influenza. A case of confirmed influenza was any patient with a positive influenza PCR assay.

Japanese authorities initially required SARS-CoV-2 positive patients to have two consecutive RT-PCR negative oropharyngeal swabs collected 24 hours apart for hospital discharge. The Australian Government Department of Foreign Affairs and Trade (DFAT) requested Japanese hospitals use the SoNG for discharge of Australian patients. On February 21 guidelines were updated to reflect the use of nasopharyngeal specimens in light of initial evidence suggesting greater viral concentrations in nasopharyngeal, compared with oropharyngeal secretions [11]. We defined the time to clear SARS-CoV-2 infection in passengers in Japan and repatriated to Australia as the period from first positive test to the first of two negative nasopharyngeal tests 24 hours apart.

Standard contingency tables supplemented by Fisher exact and chi-squared tests were done using STATA version 13.1 (StataCorp® LLC, College Station, Texas, United States). We used a log-rank test and Cox regression analysis to explore the impact of demographics and clinical risk factors on the duration of SARS-CoV-2 carriage. Univariate relative risks (RR) were calculated to determine associations between potential risk factors and SARS-CoV-2 infection. We excluded incomplete responses from the analysis. The chosen statistical significance limit was set at $p < 0.05$ with a confidence interval of 95%.

The Australian National University Human Research Ethics Committee (Protocol 2017/909) has a standing approval for outbreak investigations involving staff and students. Participants provided oral consent at the time of interview. The investigation was conducted under routine Australian public health legislation to control outbreaks of infectious disease.

## Results

### Case characteristics

Australian passengers accounted for 6% (223/3,711) of all passengers and crew on the Diamond Princess. Of these, 25% (56/223) were diagnosed SARS-CoV-2 positive by 27 February, 82% (46/56) of whom were diagnosed in Japan and 18% (10/56) after repatriation in Australia (Table 1). From the total of 223 Australian passengers, we interviewed 196 individuals with a response rate of 79% (44/56) for SARS-CoV-2 positive cases and 91% (152/167) for SARS-CoV-2 negative individuals. Six individuals declined to participate and 21 were non-contactable.

We identified that 71% (35/49) of SARS-CoV-2 cases experienced symptoms prior to virologic clearance and 97% (34/35) could identify their exact date of illness onset. Seventeen percent (6/35) had symptoms before quarantine commenced on 5 February and two-thirds (23/35) of symptomatic cases occurred within one incubation period (mean 5.2 days, 95th percentile 12.5 days [12]) of the start of the ship-based quarantine, including four people who were repatriated but identified with mild symptoms (Fig 2). The remaining 14% (5/35) of symptomatic COVID-19 cases developed their symptoms either at the repatriation facility or after transfer to their jurisdiction of residence within second the quarantine period. The mean serial interval between symptom onset in a primary case and a shared cabin close-contact displaying COVID-19 symptoms was 4.7 days (range 1–11 days).

**Table 1. Characteristics of Australian passengers on the Diamond Princess, March 2020 (n = 223).**

| | | SARS-CoV-2 positive *n* | SARS-CoV-2 negative *n* | Attack rate | *p*-value |
|---|---|---|---|---|---|
| Australian passengers | | 56 | 167 | 25% | |
| Gender | Male | 26 | 78 | 25% | 0.97 |
| | Female | 30 | 89 | 25% | |
| Age groups, years | 0–9 years | 0 | 1 | 0% | 0.05 |
| | 10–19 years | 3 | 7 | 30% | |
| | 20–29 years | 5 | 3 | 63% | |
| | 30–39 years | 0 | 5 | 0% | |
| | 40–49 years | 2 | 8 | 20% | |
| | 50–59 years | 12 | 20 | 38% | |
| | 60–69 years | 14 | 68 | 17% | |
| | 70–79 years | 19 | 46 | 29% | |
| | 80–89 years | 1 | 9 | 10% | |
| Cabin occupancy | 1 person | 0 | 5 | 0% | 0.12 |
| | 2 person | 51 | 138 | 27% | |
| | 3 person | 1 | 16 | 6% | |
| | 4 person | 4 | 8 | 33% | |

Non-specific ARI were reported by almost a third of respondents (62/196) and 84% (52/62) could identify their onset date. Thirty-seven passengers reported their onset while on board the Diamond Princess, while another 15 after arrival in quarantine in Australia. Five individuals were confirmed influenza A positive, including one individual who tested influenza A positive and a fortnight later SARS-CoV-2 positive. Seventy-one percent (25/35) of symptomatic COVID-19 cases reported experiencing fever ($\geq$37.5˚C) compared to 23% (14/62) of people reporting non-specific ARI ($p$<0.001, Table 2). Twenty percent of COVID-19 cases also had diarrhoea (7/35) and six reported other signs and symptoms not specifically captured in the questionnaire, including three with tight chest/chest pains, two with loss of smell and taste and one with chills, but no documented fever.

All SARS-CoV-2 cases were hospitalised on diagnosis, irrespective of illness severity. After repatriation, one passenger died of COVID-19 induced acute respiratory distress before the

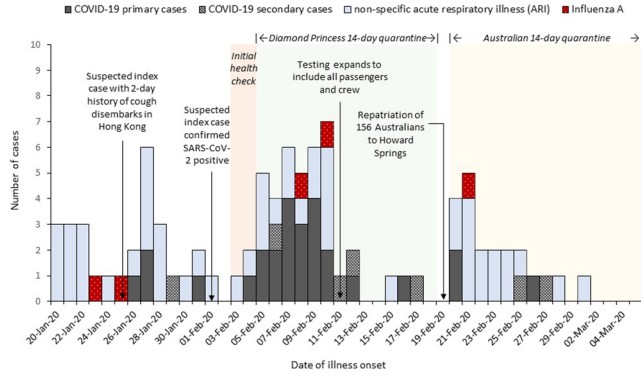

**Fig 2. Epidemic curve of Australian symptomatic cases by date of COVID-19 illness onset (n = 34), non-specific acute respiratory illness (n = 52) and influenza A (n = 5) on the Diamond Princess and after repatriation, 20 January to 5 March 2020.**

**Table 2. Signs and symptoms of Australian passengers diagnosed with COVID-19 disease (n = 35) and non-specific ARI (n = 62), excluding influenza.**

|  |  | COVID-19 cases (*n* = 35) | | | non-specific ARI (*n* = 62) | | | *p*-value |
|---|---|---|---|---|---|---|---|---|
|  |  | Number reporting (%) | Median duration, days | Range, days | Number reporting (%) | Median duration, days | Range, days |  |
| Signs and symptoms | Fever | 25 (71) | 3 | 1–10 | 14 (23) | 1.5 | 1–11 | <0.001* |
|  | Cough | 13 (37) | 5 | 1->14 | 29 (47) | 3 | 1->14 | 0.36 |
|  | Headache | 9 (26) | 2 | 1–3 | 16 (26) | 3 | 1->14 | 0.99 |
|  | Sore throat | 9 (26) | 2 | 1–12 | 21 (34) | 2 | 1–11 | 0.40 |
|  | Runny nose | 8 (23) | 2.5 | 1–12 | 7 (11) | 4 | 1–10 | 0.13 |
|  | Unusual fatigue | 7 (20) | 3 | 1->14 | 6 (10) | 3 | 2->14 | 0.15 |
|  | Shortness of breath | 3 (9) | 5 | 3–10 | 6 (10) | 1.5 | 1–3 | 1.00 |

*Statistically significant.

>14 days indicates the symptom was still experienced at the time of the questionnaire.

study commenced. Of the cases diagnosed in Japan, a chest x-ray was performed on 70% (25/36) and a chest computed tomography (CT) scan on 47% (17/36). One-third of cases managed in Japan received both an x-ray and a CT scan (12/36). Only 50% (4/8) of cases managed in Australia had a chest x-ray, and none a chest CT. Of all SARS-CoV-2 respondents, eight participants (18%, 8/44) reported that their radiological investigations confirmed pneumonia.

### Length of time to clear SARS-CoV-2 infection

We found 29% (14/49) of SARS-CoV-2 infections were persistently asymptomatic and we obtained virologic clearance data for 37% (18/49) of the cases. The median time-to-clearance of SARS-CoV-2 infection was shorter for asymptomatic cases (13 days, range 5–17 days) compared with symptomatic cases (19 days, range 10–22 days), *p* = 0.002 (Fig 3). While symptom status was a statistically significant predictor of time-to-clearance, the binary variable age group (>65 or ≤65 years) was not, but we retained it in the prediction model due to biological plausibility. Overall, the adjusted hazard ratio for virologic clearance in symptomatic versus asymptomatic patients was 0.14 (95% CI 0.03–0.65), suggesting that asymptomatic patients were 86% more likely to have cleared their infection at any particular time point during follow-up.

### Risk factors

People sharing a cabin or who were a close contact of a confirmed SARS-CoV-2 case prior to ship quarantine commencing were 3.78 (95% CI 2.24–6.37) times more likely to become infected compared to those with no known exposure (Table 3). The relative risk of testing SARS-CoV-2 positive from an exposure to a known SARS-CoV-2 positive cabin mate during ship-based quarantine was 6.18 (95% CI 1.96–19.46). There was no statistically significant association between attending shore trips, touring in large groups, participating in social events before quarantine or visiting public areas during quarantine and subsequent SARS-CoV-2 infection.

### Discussion

This outbreak of COVID-19 on board the Diamond Princess cruise ship was the first major outbreak involving Australians and was the largest outbreak outside of Wuhan, China at the

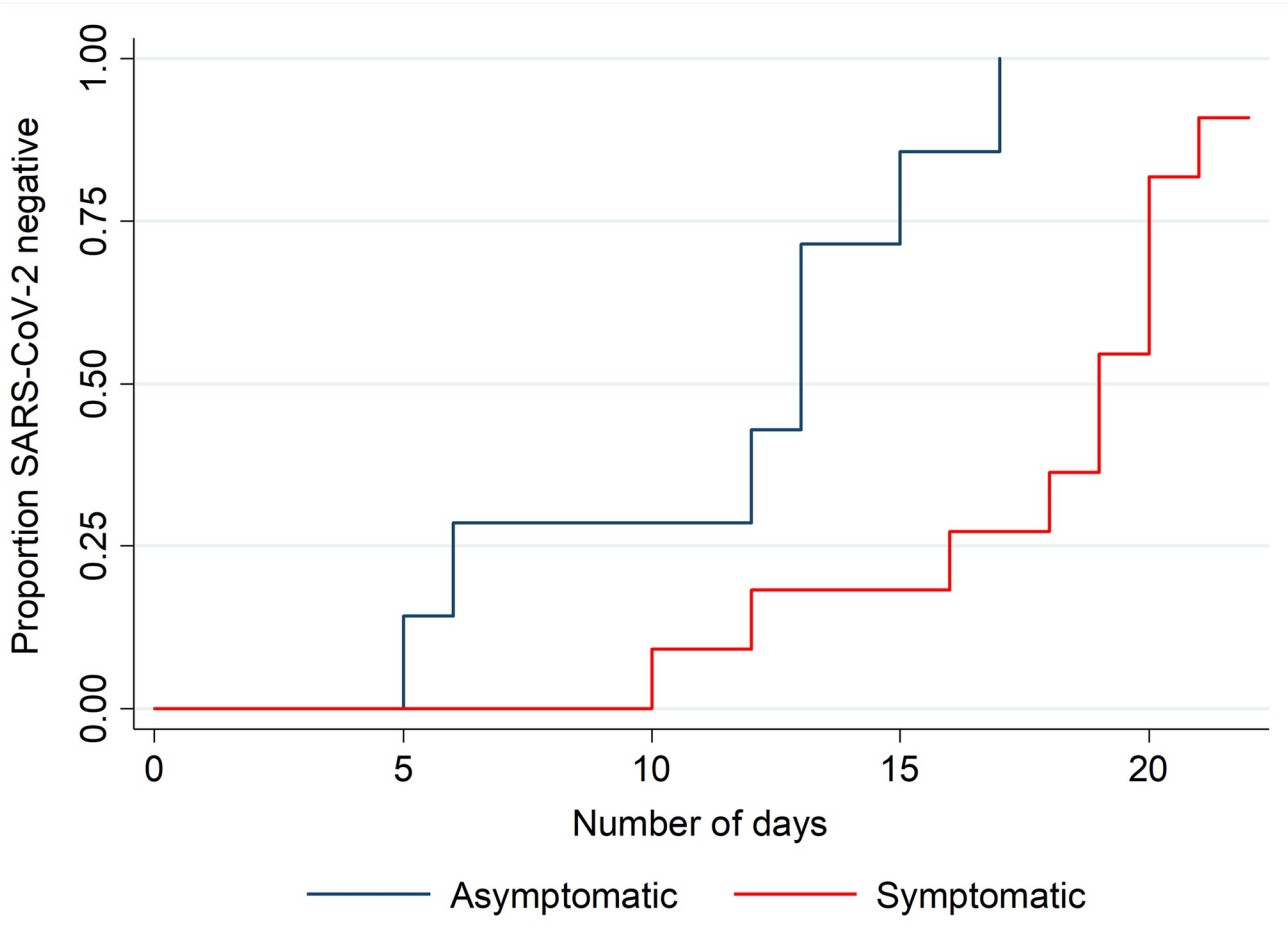

**Fig 3. Cumulative proportion virologically cleared by symptom status and time of follow-up (n = 18), March 2020.**

time. A quarter of the Australian passengers on the vessel tested positive for SARS-CoV-2, similar to the rate of infection of the entire Diamond Princess cohort of 19.2% [13]. Of the SARS--CoV-2 positive Australians, almost a third were persistently asymptomatic, varying from the 46.5% of all Diamond Princess positive cases at time of testing [13, 14]. Two further studies indicated 19–23% developed symptoms during follow-up and could be reclassified as pre-symptomatic [15, 16]. Our similar observations of a lower proportion of asymptomatic cases is possible because we enquired about a wide spectrum of symptoms during interview and a small number of cases subsequently developed symptoms after their initial test. Asymptomatic testing was not conducted after repatriation however all repatriates completed a further 14 days of quarantine in an Australian facility with sensitive daily screening for symptoms and temperature checks. This may have biased our findings to symptomatic individuals, which would have biased our results toward the null. The epidemic curve shows SARS-CoV-2 infections were occurring amongst Australians before ship-based quarantine commenced. In this cohort, the illness peaked around 3–5 days after quarantine started which supports previous findings that the movement restrictions placed on 5 February reduced the risk of infection among those passengers who had no known close contact with an infected individual [17, 18]. While passenger-to-passenger transmission, the dominant transmission type, decreased with the introduction of movement restrictions, crew-based transmission remained steady later

**Table 3. Risk factors for SARS-CoV-2 infection among Australian passengers on the Diamond Princess cruise ship, March 2020.**

| Risk factors & exposures | | SARS-CoV-2 positive | | | SARS-CoV-2 negative | | | |
|---|---|---|---|---|---|---|---|---|
| | | Exp | Non-exp | AR % | Exp | Non-exp | AR % | RR (95% CI) |
| Demographics | Sex—male | 27 | 22 | 55 | 86 | 66 | 57 | 0.96 (0.59–1.56) |
| | Age ≥65 years | 21 | 28 | 43 | 84 | 68 | 55 | 0.69 (0.42–1.12) |
| Shore trips | Kagoshima, Japan 22 Jan | 41 | 3 | 93 | 130 | 22 | 86 | 2.00 (0.67–5.97) |
| | Hong Kong, China 25 Jan | 37 | 7 | 84 | 132 | 20 | 87 | 0.84 (0.42–1.70) |
| | Chan May, Vietnam 27 Jan | 41 | 3 | 93 | 133 | 19 | 88 | 1.73 (0.58–5.11) |
| | Cai Lan, Vietnam 28 Jan | 40 | 4 | 91 | 140 | 12 | 92 | 0.89 (0.36–2.17) |
| | Keelung, Taiwan 31 Jan | 41 | 3 | 93 | 132 | 20 | 87 | 1.81 (0.61–5.40) |
| | Okinawa, Japan 1 Feb | 41 | 3 | 93 | 119 | 33 | 78 | 3.07 (1.01–9.38)* |
| | Group tours ≥15 people | 34 | 10 | 77 | 120 | 32 | 79 | 0.93 (0.50–1.72) |
| Visited public areas | Attended social events 3–4 Feb | 24 | 17 | 59 | 100 | 50 | 67 | 0.76 (0.44–1.32) |
| | Week 1 of quarantine | 20 | 16 | 56 | 86 | 66 | 57 | 0.98 (0.55–1.78) |
| | Week 2 of quarantine | 13 | 14 | 48 | 86 | 62 | 58 | 0.71 (0.36–1.43) |
| Exposure to a confirmed case | Before ship quarantine, cabin mate or close contact | 33 | 16 | 67 | 38 | 114 | 25 | 3.78 (2.24–6.37)* |
| | During ship quarantine, cabin mate | 4 | 6 | 40 | 11 | 133 | 8% | 6.18 (1.96–19.46)* |

Exp = exposure, AR = attack rate, RR = relative risk

*statistically significant.

into the quarantine period, likely due to continued movement of crew providing services for passengers [19].

Half of the SARS-CoV-2 positive cases identified by the Japanese authorities at initial testing were asymptomatic [14]. Household and close contact studies of COVID-19 have found asymptomatic cases can result in person-to-person transmission [4, 20]. Based on the discharge guidelines observed at the time of the study, we found that asymptomatic cases remained SARS-CoV-2 positive for a median of 13-days. Other studies have found a comparable duration for virological carriage, although asymptomatic cases in these series have typically been younger [15, 20–23]. While positive RT-PCR results imply potential infectivity, viable SARS-CoV-2 has been isolated from asymptomatic and pre-symptomatic cases [23]. Results of viral culture suggest that the majority of COVID-19 cases do not shed infectious SARS-CoV-2 virus beyond 8 days from symptom onset even in the context of on-going PCR-positive specimens [24]. The expected duration of infectivity for asymptomatic cases remains unclear.

Environmental sampling of passenger and crew cabins following disembarkation found viral fragments in cabins of both symptomatic (15% of samples) and asymptomatic (21% of samples) cases, with the most common locations being pillows and toilet floors [30]. This suggests that environmental surfaces may be involved in transmission, particular in regularly use spaces, such as bathrooms and beds, and highlights the role of asymptomatic infections in spreading disease [25]. No viable virus was located in this sampling process.

Extensive socialising of passengers and the propagated person-to-person transmission of SARS-CoV-2 likely affected our ability to identify specific exposure activities that increased the risk of SARS-CoV-2 infection. Although we did find the risk of infection was higher if an individual shared a cabin or was a close contact with a confirmed SARS-CoV-2 case. This is consistent with findings from other studies, with the attack rate among cabinmates on the Diamond Princess between 56% and 83% far higher than that found in general households of 11.2% to 19.3%, likely due to persistent and sustained exposure in the confined cabin space [26–28]. Based on the challenges of ship-based quarantine, the identification of additional cases after

repatriation was not unexpected. These ten cases, while negative when tested by Japanese authorities and afebrile at time of repatriation, could be explained by an inadequate collected specimen or as speculated by a Hong Kong repatriation study, a specimen collected in the incubation stage of their infection and undetectable by RT-PCR at the time [29]. Cases with no link to a confirmed cabin mate may have been infected by crew who were unable to effectively quarantine while performing food delivery or from fomites (eating utensils and trays) before onshore catering supplied food in disposable packaging [18, 30]. The onset of illness in the first crew member began on 2 February and by 9 February food service workers represented the largest proportion of febrile crew members [31]. This highlights the critical requirement for a comprehensive outbreak plan that includes mitigating risk for crew and food handling and delivery protocols.

Australian passengers accounted for just 6% of all individuals on board and the small sample size limits the statistical power of our study. The response rate (79%) from SARS-CoV-2 cases might indicate non-responders had more severe COVID-19 disease, however we are unaware of any Australians who would fall into this group. Length of time to clear infection was obtained from patients' personal diaries rather than test results, but it was likely to be accurate given that information related to their discharge from hospital. While we were only able to obtain virological clearance data on 37% of cases, we believe this information would be broadly representative of all cases.

At the time of the Diamond Princess outbreak, little was known about novel SARS-CoV-2 natural history and public health response protocols for COVID-19 on board ships were unavailable. Managing any respiratory disease on a vessel with a high-density population using shared facilities is challenging. The high number of asymptomatic cases and potential for severe COVID-19 disease in older adults requires enhanced screening and the use of COVID-19 point of care test kits for rapid detection. Heightened sanitation measures should become routine in public areas and cabins, and the vessel should be supplied with sufficient amounts of appropriate personal protective equipment for both passengers and crew. To reduce the handling of food, catering activities during an outbreak should favour individually packaged meals delivered under strict hygiene protocols to avoid contamination and transmission. Easily accessible and strategically positioned hand hygiene stations should be a standard feature in all cabins and throughout the ship, particularly at entrances to food and beverage outlets. Frequent handwashing and hygiene etiquette can be reinforced through daily broadcasted health messages. If suspected cases are identified, routine infection control practice should include prompt isolation in single cabins, without shared facilities. As the average length of many cruises is less than 14-days, vessels should be guaranteed port entry to disembark infected individuals and their close contacts to onshore isolation and quarantine facilities, as is now recommended in the interim guidance from the World Health Organization [32, 33]. The identification, preparation and use of such onshore facilities, as well as the safe transfer of passengers to them, will require a multi-agency collaboration at the local, jurisdictional and national level. The results of this investigation show the highly infectious nature of SARS-CoV-2 on cruise ships and the consequences in an older vulnerable population. In the future, non-pharmaceutical interventions along with vaccination of travellers will prove important requirements for resumption of cruises.

## Conclusions

Like many countries, Australia has extended its ban on cruise ships entering its territorial waters in order to establish adequate maritime specific COVID-19 outbreak management plans and to ensure the health and safety of the broader community. An international

guidance document to assist cruise ship companies in the development of robust, standardised and auditable public health plans for their operations should be considered.

## Acknowledgments

We thank members of the Australian Medical Assistance Team (AUSMAT), the National Critical Care and Trauma Response Centre (NCCTRC) and the National Incident Room who conducted the repatriation, quarantine and interviewing of the passengers. We also thank the research officers at the National Centre for Epidemiology & Population Health who entered data into REDCap and the Japanese National Focal Point for providing the Diamond Princess manifest of Australian nationals and in-country testing outcomes.

We especially thank the Australian passengers of the Diamond Princess without whose cooperation this investigation could not have been conducted.

## Author Contributions

**Conceptualization:** Paul Kelly, Martyn D. Kirk.

**Data curation:** Liz J. Walker, Samantha M. Colquhoun.

**Formal analysis:** Liz J. Walker.

**Investigation:** Liz J. Walker, Tudor A. Codreanu, Paul K. Armstrong, Sam Goodwin, Abigail Trewin, Emma Spencer, Dianne M. Stephens, Rob W. Baird, Nicholas M. Douglas.

**Supervision:** Martyn D. Kirk.

**Writing – original draft:** Liz J. Walker, Tudor A. Codreanu.

**Writing – review & editing:** Liz J. Walker, Tudor A. Codreanu, Paul K. Armstrong, Sam Goodwin, Abigail Trewin, Emma Spencer, Samantha M. Colquhoun, Dianne M. Stephens, Rob W. Baird, Nicholas M. Douglas, Danielle Cribb, Rhonda Owen, Paul Kelly, Martyn D. Kirk.

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
