## [Decision Letter · Decision Letter 0]

30 Mar 2021

PONE-D-20-38119

SARS-CoV-2 infections among Australian passengers on the Diamond Princess cruise ship: a retrospective cohort study.

PLOS ONE

Dear Dr. Walker,

Thank you for submitting your manuscript to PLOS ONE. After careful consideration, we feel that it has merit but does not fully meet PLOS ONE’s publication criteria as it currently stands. Therefore, we invite you to submit a revised version of the manuscript that addresses the points raised during the review process.

The Authors are expected to address all the criticisms by all Reviewers. In particular, please provide the rationale of the modelling strategy, potential impact of non-response (Reviewers #1), the impact of asymptomatic infections in the results and conclusion, and further clarify the timing and type of actions carried out in Japan or Australia (Reviewer #2). In additional to the above comments, please address,

Could the authors add more discussion on how the results will inform infection control policy while we expect cruise ship is likely to resume in the near future, where various non-pharmaceutical interventions are now more widely accepted and some passengers are vaccinated.Line 179-180. “The median time-to-clearance of SARS-CoV-2 infection was lower for asymptomatic cases.” Please replace the word “lower” with “shorter”.Table 2. Range, please clarify the meaning of “1 -> 14”

We look forward to receiving your revised manuscript.

Kind regards,

Eric HY Lau, Ph.D.

Academic Editor

PLOS ONE

Journal Requirements:

"The Australian National University Human Research Ethics Committee (Protocol 2017/909) has a standing approval for outbreak investigations involving staff and students. The investigation was consistent with routine Australian public health activities during outbreaks of infectious disease.".   

Please provide additional details regarding participant consent.

As your study also involves a retrospective analysis  of medical records or archived samples, please ensure that you have discussed whether all data were fully anonymized before you accessed them and/or whether the IRB or ethics committee waived the requirement for informed consent. If patients provided informed written consent to have data from their medical records used in research, please include this information.

Once you have amended this statement in the Methods section of the manuscript, please add the same text to the “Ethics Statement” field of the submission form (via “Edit Submission”).

Additional Editor Comments (if provided):

The Authors are expected to address all the criticisms by all Reviewers. In particular, please provide the rationale of the modelling strategy, potential impact of non-response (Reviewers #1), the impact of asymptomatic infections in the results and conclusion, and further clarify the timing and type of actions carried out in Japan or Australia (Reviewer #2). In additional to the above comments, please address,

1. Could the authors add more discussion on how the results will inform infection control policy while we expect cruise ship is likely to resume in the near future, where various non-pharmaceutical interventions are now more widely accepted and some passengers are vaccinated.

2. Line 179-180. “The median time-to-clearance of SARS-CoV-2 infection was lower for asymptomatic cases.” Please replace the word “lower” with “shorter”.

3. Table 2. Range, please clarify the meaning of “1 -> 14”

Reviewers' comments:

Reviewer's Responses to Questions

**Comments to the Author**

1. Is the manuscript technically sound, and do the data support the conclusions?

Reviewer #1: Yes

Reviewer #2: Partly

2. Has the statistical analysis been performed appropriately and rigorously? 

Reviewer #1: Yes

Reviewer #2: Yes

3. Have the authors made all data underlying the findings in their manuscript fully available?

Reviewer #1: Yes

Reviewer #2: Yes

4. Is the manuscript presented in an intelligible fashion and written in standard English?

Reviewer #1: Yes

Reviewer #2: Yes

5. Review Comments to the Author

Reviewer #1: This is a retrospective cohort study about Australian passengers on the Diamond Princess which shows the natural history of COVID-19. This study highlighted challenges of cruise ship-related COVID-19 outbreak and the tips they learned are worth publishing. However, I have some comments to improve the manuscript before publication.

Major comments:

1. The authors conducted Cox proportional hazards regression analysis for the prediction of time-to-clearance of SARS-CoV-2, but little is discussed or shown about the model. This explanation will facilitate readers to understand the findings of the study, including why the authors categorize the age as a binary variable.

2. Age, gender and race may influence the development of COVID-19. It is better to describe their distribution among those who censored (who did not respond to the questionnaire), if the information is available.

Minor comment:

Line 226. “Viral culture suggest” → “Results of viral culture suggest”.

Reviewer #2: The authors presented the retrospective cohort data from Australian passengers in the Diamond Princess outbreak. The authors characterized and classified the passengers into groups and determined the factors that could affect COVID-19 transmissibility and infectivity. It is clear that the information presented in the manuscript is important to the control of COVID-19. Nevertheless, the event happened in the early period of the pandemic. The epidemic and observational data in the manuscript have been thoroughly covered and discussed in many COVID-19 articles.

Areas for improvement:

1. Considering the available published papers on the Diamond Princess cohort, the Introduction and Discussion sections could be significantly improved by citing and discussing those findings.

2. Line 105: the lack of RT-PCR testing in the repatriated group without known symptoms is a major concern since the majority of the infected individuals might not be symptomatic. This could mean that the findings were biased toward symptomatic cases only, which would negate all the conclusions. It will be helpful for the authors to address this point and rewrite the manuscript accordingly.

3. Line 178: it appears here that there is the data from asymptomatic infected individuals. This is in contrary to point #2 that RT-PCR testing was not done in this group. Is the data from Japan or obtained after repatriation? Please also list the criteria for clearance used in Australia at that time.

4. The Discussion section has several points on the best practice for COVID-19 prevention (examples in Line 257, 258 and 262). However, they are not related to the data shown in the manuscript. If they are presented, the data leading to these points will be needed.

5. In the Methods section, it is difficult to fully understand which activities were done in Australia or in Japan and when they were performed. The authors can clarify them in the text or introduce the information in either Figure 1 or Figure 2.

6. Line 89: the name of the document is incorrect. Please use its full name.

6. PLOS authors have the option to publish the peer review history of their article (what does this mean?). If published, this will include your full peer review and any attached files.

Reviewer #1: No

Reviewer #2: No

---

## [Author Response · Author response to Decision Letter 0]

23 May 2021

Reviewer 1 comments:

1. The authors conducted Cox proportional hazards regression analysis for the prediction of time-to-clearance of SARS-CoV-2, but little is discussed or shown about the model. This explanation will facilitate readers to understand the findings of the study, including why the authors categorize the age as a binary variable. 

Response: This was a very simple survival analysis to account for censoring. Evidence at the time of the outbreak indicated older age may have been a risk factor for infection and longer shedding. Due to the small sample size we were unable to separate age into broader age group, so elected for a binary age group category. Our model found age was not statistically significant and this is likely due to the intensive social mixing that occurs on a cruise ship.

2. Age, gender and race may influence the development of COVID-19. It is better to describe their distribution among those who censored (who did not respond to the questionnaire), if the information is available. 

Response: Unfortunately, we do not know the date of symptom onset for individuals who were unable to be contacted or declined participation, therefore we are unable to include all positive SARS-CoV-2 Australian cases in the Cox regression.

Our response rate from SARS-CoV-2 cases was high (78%) and we believe the current model would not differ much if we had data on the 12 censored individuals.

3. Line 226. “Viral culture suggest” → “Results of viral culture suggest”. 

Response: Sentence has been amended to ‘Results of viral culture suggest…’ 

Reviewer 2 comments:

1. Considering the available published papers on the Diamond Princess cohort, the Introduction and Discussion sections could be significantly improved by citing and discussing those findings. 

Response: A literature review was conducted on 29 April 2021 using search term Diamond Princess. The introduction and discussion have been updated to incorporate findings from other studies of the Diamond Princess cohort.

2. Line 105: the lack of RT-PCR testing in the repatriated group without known symptoms is a major concern since the majority of the infected individuals might not be symptomatic. This could mean that the findings were biased toward symptomatic cases only, which would negate all the conclusions. It will be helpful for the authors to address this point and rewrite the manuscript accordingly.

Response: At the time of this outbreak, routine screening was not conducted of the entire cohort that was repatriated. However, all passengers were screened on board the Diamond Princess prior to their repatriation. At the quarantine facility, there was a sensitive screening system based on daily temperature checks and symptoms. There may have been a bias towards toward symptomatic cases. However, this is normal in outbreak investigations for infectious agents where screening of entire cohorts is rare. Our findings and conclusions would be biased toward the null. We have noted this in the discussion.

3. Line 178: it appears here that there is the data from asymptomatic infected individuals. This is incontrary to point #2 that RT-PCR testing was not done in this group. Is the data from Japan or obtained after repatriation? Please also list the criteria for clearance used in Australia at that time. 

Response: As we mentioned in the manuscript, all passengers were screened for SARS-CoV-2 on board the Diamond Princess. The criteria for clearance are listed in the methods. We have added text to clarify that this applied to both passengers in Japan and Australia.

4. The Discussion section has several points on the best practice for COVID-19 prevention (examples in Line 257, 258 and 262). However, they are not related to the data shown in the manuscript. If they are presented, the data leading to these points will be needed.

Response: We disagree, these are our reflections on the findings from the experience of investigating this outbreak. These are public health practices that prevent infections on board cruise ships. Where possible, we have referenced our statements.

5. In the Methods section, it is difficult to fully understand which activities were done in Australia or in Japan and when they were performed. The authors can clarify them in the text or introduce the information in either Figure 1 or Figure 2.

Response: The majority of work for this investigation was conducted in Australia. The exceptions were the laboratory work, where some of this occurred in Japan, which is identified. We have noted in the manuscript that interviews were conducted on-site at the quarantine facility or via telephone for people remaining in Japan.

6. Line 89: the name of the document is incorrect. Please use its full name.

Response: The name of the document has been correct to ‘The first few X cases and contacts (FFX) investigation protocol for coronavirus disease 2019 (COVID-19)‎’.

Editors comments:

1. Could the authors add more discussion on how the results will inform infection control policy while we expect cruise ship is likely to resume in the near future, where various nonpharmaceutical interventions are now more widely accepted, and some passengers are vaccinated.

Response: We have added the following paragraph to the final part of the discussion:

‘The results of this investigation show the highly infectious nature of SARS-CoV-2 on cruise ships and the consequences in an older vulnerable population. In the future, non-pharmaceutical interventions along with vaccination of travellers will prove important requirments for resumption of cruises.’

2. Line 179-180. “The median time-to- clearance of SARS-CoV-2 infection was lower for asymptomatic cases.” Please replace the word “lower” with “shorter”.

Response: The sentence has been amended to read ‘The median time-to- clearance of SARS-CoV-2 infection was shorter for asymptomatic cases.’

3. Table 2. Range, please clarify the meaning of “1 -> 14” 

Response: >14 days indicates the symptom was still experienced at the time of the questionnaire. A footnote has been added to Table 2.

4. Please provide additional details regarding participant consent. As your study also involves a retrospective analysis of medical records or archived samples, please ensure that you have discussed whether all data were fully anonymized before you accessed them and/or whether the IRB or ethics committee waived the requirement for informed consent. If patients provided informed written consent to have data from their medical records used in research, please include this information. 

Once you have amended this statement in the Methods section of the manuscript, please add the same text to the “Ethics Statement” field of the submission form (via “Edit Submission”).

Response: This paper details the Australian response to an outbreak of SARS-CoV-2 conducted under public health legislation. We have detailed that in manuscript.

SARS-CoV-2 pathology results conducted in Japan were obtained under the International Health Regulations (2005) between the Japanese IHR focal point to the Australian IHR focal point. SARS-CoV-2 pathology results conducted in Australia were obtained under the National Health Security Act (2007), states and territories have agreed data sharing for the national surveillance of diseases on the National Notifiable Disease List (NNDL) with the Australian Government. 

All investigators participated in this as part of their role as public health professionals and operated under public health legislation. The lead author was a Master of Applied Epidemiology scholar at the Australian National University at the time of this investigation. The Australian National University has a standing approval for MAE scholars to investigate outbreaks and evaluate surveillance, which is already documented in the manuscript. Participation was voluntary and participants provided verbal consent at the time of the survey, which we have detailed in the manuscript and the ‘ethics statement’. No medical records were obtained, information regarding medical treatment (x-ray, CT-scans) were obtained verbally from each case’s recall.

Response:We have deleted the sentences that references ‘data not shown’ from the manuscript.

---

## [Decision Letter · Decision Letter 1]

16 Jul 2021

SARS-CoV-2 infections among Australian passengers on the Diamond Princess cruise ship: a retrospective cohort study.

PONE-D-20-38119R1

Dear Dr. Walker,

We’re pleased to inform you that your manuscript has been judged scientifically suitable for publication and will be formally accepted for publication once it meets all outstanding technical requirements.

Kind regards,

Eric HY Lau, Ph.D.

Academic Editor

PLOS ONE

Additional Editor Comments (optional):

Reviewers' comments:

Reviewer's Responses to Questions

**Comments to the Author**

1. If the authors have adequately addressed your comments raised in a previous round of review and you feel that this manuscript is now acceptable for publication, you may indicate that here to bypass the “Comments to the Author” section, enter your conflict of interest statement in the “Confidential to Editor” section, and submit your "Accept" recommendation.

Reviewer #1: All comments have been addressed

Reviewer #2: All comments have been addressed

2. Is the manuscript technically sound, and do the data support the conclusions?

Reviewer #1: Yes

Reviewer #2: Yes

3. Has the statistical analysis been performed appropriately and rigorously? 

Reviewer #1: Yes

Reviewer #2: Yes

4. Have the authors made all data underlying the findings in their manuscript fully available?

Reviewer #1: Yes

Reviewer #2: Yes

5. Is the manuscript presented in an intelligible fashion and written in standard English?

Reviewer #1: Yes

Reviewer #2: Yes

6. Review Comments to the Author

Reviewer #1: (No Response)

Reviewer #2: The manuscript has been satisfactorily revised. I would like to thank the authors for conducting this important study.

7. PLOS authors have the option to publish the peer review history of their article (what does this mean?). If published, this will include your full peer review and any attached files.

Reviewer #1: No

Reviewer #2: No

---

## [Editor Report · Acceptance letter]

27 Aug 2021

PONE-D-20-38119R1 

SARS-CoV-2 infections among Australian passengers on the Diamond Princess cruise ship: a retrospective cohort study. 

Dear Dr. Walker:

I'm pleased to inform you that your manuscript has been deemed suitable for publication in PLOS ONE. Congratulations! Your manuscript is now with our production department. 

Kind regards, 

on behalf of

Dr. Eric HY Lau 

Academic Editor

PLOS ONE